# Single-Breath-Hold MRI-SPACE Cholangiopancreatography with Compressed Sensing versus Conventional Respiratory-Triggered MRI-SPACE Cholangiopancreatography at 3Tesla: Comparison of Image Quality and Diagnostic Confidence

**DOI:** 10.3390/diagnostics11101886

**Published:** 2021-10-13

**Authors:** Olivier Chevallier, Hélène Escande, Khalid Ambarki, Elisabeth Weiland, Bernd Kuehn, Kévin Guillen, Sylvain Manfredi, Sophie Gehin, Julie Blanc, Romaric Loffroy

**Affiliations:** 1Department of Vascular and Interventional Radiology, Image-Guided Therapy Center, François-Mitterrand University Hospital, 14 Rue Paul Gaffarel, BP 77908, 21079 Dijon, France; olivier.chevallier@chu-dijon.fr (O.C.); escande.helene@hotmail.fr (H.E.); kevin.guillen@chu-dijon.fr (K.G.); sophie.gehin@chu-dijon.fr (S.G.); 2Siemens Healthcare SAS, 97715 Paris, France; khalid.ambarki@siemens-healthineers.com; 3MR Application Predevelopment, Siemens Healthcare GmbH, 91054 Erlangen, Germany; elisabeth.weiland@siemens-healthineers.com (E.W.); bernd.kuehn@siemens-healthineers.com (B.K.); 4Department of Gastroenterology and Hepatology, François-Mitterrand University Hospital, 14 Rue Paul Gaffarel, BP 77908, 21079 Dijon, France; sylvain.manfredi@chu-dijon.fr; 5Department of Biostatistics, Georges-François Leclerc Cancer Center, 1 Rue du Professeur Marion, 21000 Dijon, France; jblanc@cgfl.fr

**Keywords:** MRI, cholangiography, cholangiopancreatography, MRCP, bile, compressed sensing, lithiasis, IPMN

## Abstract

To compare two magnetic resonance cholangiopancreatography (MRCP) sequences at 3 Tesla (3T): the conventional 3D Respiratory-Triggered SPACE sequence (RT-MRCP) and a prototype 3D Compressed-Sensing Breath-Hold SPACE sequence (CS-BH-MRCP), in terms of qualitative and quantitative image quality and radiologist’s diagnostic confidence for detecting common bile duct (CBD) lithiasis, biliary anastomosis stenosis in liver-transplant recipients, and communication of pancreatic cyst with the main pancreatic duct (MPD). Sixty-eight patients with suspicion of choledocholithiasis or biliary anastomosis stenosis after liver transplant, or branch-duct intraductal papillary mucinous neoplasm of the pancreas (BD-IPMN), were included. The relative CBD to peri-biliary tissues (PBT) contrast ratio (CR) was assessed. Overall image quality, presence of artefacts, background noise suppression and the visualization of 12 separated segments of the pancreatic and bile ducts were evaluated by two observers working independently on a five-point scale. Diagnostic confidence was scored on a 1–3 scale. The CS-BH-MRCP presented significantly better CRs (*p* < 0.0001), image quality (*p* = 0.004), background noise suppression (*p* = 0.011), fewer artefacts (*p* = 0.004) and better visualization of pancreatic and bile ducts segments with the exception of the proximal CBD (*p* = 0.054), cystic duct confluence (*p* = 0.459), the four secondary intrahepatic bile ducts, and central part of the MPD (*p* = 0.885) for which no significant differences were found. Overall, diagnostic confidence was significantly better with the CS-BH-MRCP sequence for both readers (*p* = 0.038 and *p* = 0.038, respectively). This study shows that the CS-BH-MRCP sequence presents overall better image quality and bile and pancreatic ducts visualization compared to the conventional RT-MRCP sequence at 3T.

## 1. Introduction

Magnetic resonance cholangiopancreatography (MRCP) is a non-invasive imaging method used in everyday clinical practice to assess anatomical features and abnormalities of the intrahepatic and extrahepatic bile ducts and of the pancreatic ducts [1,2,3,4,5]. MRCP does not involve the risks associated with endoscopic retrograde cholangiopancreatography (ERCP) such as acute pancreatitis, bowel perforation, infections, and bleeding [6]. Recent years have witnessed the development of three-dimensional (3D) imaging, which provides better image quality and greater diagnostic confidence compared to two-dimensional (2D) imaging [7].

Two 3D MRCP breathing management methods are currently available. The free-breathing (FB) method relies on respiratory gating has well-established diagnostic performance and spatial resolution characteristics when used to evaluate diseases of the bile ducts and pancreas [2]. However, this method requires quite long acquisition time. The other method, in which the images are acquired during a single breath-hold (BH), is being actively developed.

Research into means of improving MRCP sequences has several goals, of which the most important is reduction in the acquisition time in order to limit motion artifacts, notably those due to breathing, which are particularly challenging when imaging the abdomen. A shorter acquisition time also improves the comfort of the patient, who must remain immobile throughout the acquisition. Another advantage is the imaging of a greater number of patients during a given time, which decreases costs and wait-list times.

In practice, the acquisition time and diagnostic quality of FB MRCP with respiratory triggering (RT) are variable and difficult to predict. Only part of the k-space is acquired during each breathing cycle. Diagnostic performance may be adversely affected by irregular breathing due, for instance, to abdominal pain or failure of the machine to detect breaths. However, irregular breathing can also occur in patients with a World Health Organization (WHO) performance status of 0 for unknown reasons [8]. FB with RT acquisition times of 7 min [9], and 6 min [10], have been reported. Longer acquisition times are often associated with poorer image quality [9,11].

3D MRCP is obtained using a T2-weighted fast spin-echo sequence with the variable-flip-angle technique, such as SPACE (Sampling Perfection with Application-optimized Contrast using different flip-angle Evolutions), CUBE, and VISTA (Volume Isotropic Turbo spin echo Acquisition). The high flip angles of radiofrequency pulses for the lines near the center of the Fourier space ensure a good contrast-to-noise ratio (CNR), while the lower angles for the lines at the periphery of the Fourier space provide good spatial resolution. The clinical relevance of the SPACE technique for MRCP has been demonstrated [12].

Methods that have been evaluated to decrease the acquisition time include parallel imaging (PI) and compressed sensing (CS), which combines data undersampling with iterative reconstruction. PI has entered the mainstream of clinical practice. The decrease in acquisition time is achieved by undersampling the k-space; for instance, by skipping every other line. The main drawbacks are the moderate acceleration factors in the 2–4 range and a decrease in the CNR [13,14]. CS has undergone considerable development in recent years as a method for imaging moving targets, such as the heart and abdominal organs [15,16,17,18,19]. CS relies on the sparsity of redundant data within standard magnetic resonance images. Random undersampling of the compressible representation of an image is performed, and the image is then restored by iterative reconstruction [20].

Several studies have evaluated the clinical feasibility of CS with FB or BH acquisition, with interesting results that favored BH sequences [9,21,22,23]. A prototype CS-BH-SPACE MRCP sequence produced by Siemens Healthcare (Erlangen, Germany) seeks to eliminate motion artifacts during a BH.

The objective of this study was to prospectively compare two MRCP sequences at 3T, namely, the conventional 3D RT-SPACE sequence (designated RT-MRCP hereafter) and the prototype 3D CS-BH-SPACE sequence (designated CS-BH-MRCP hereafter), in terms of qualitative and quantitative image quality and radiologist’s diagnostic confidence for detecting common bile duct (CBD) lithiasis, biliary anastomosis stenosis in liver-transplant recipients, and communication of pancreatic cyst with the main pancreatic duct (MPD), the latter allowing the non-invasive diagnosis of branch-duct intraductal papillary mucinous neoplasm of the pancreas (BD-IPMN).

## 2. Materials and Methods

### 2.1. Study Population

This prospective single-center study included patients who underwent 3T MRCP between September 2018 and August 2020 in our institution. Consecutive patients older than 18 years of age who had clinical and laboratory-test abnormalities suggestive of choledocholithiasis or biliary anastomosis stenosis after liver transplant, or required MR imaging for cystic pancreatic lesion characterization, were included. Exclusion criteria were contraindications to MRI, moderate-to-abundant ascites, an inability to maintain a breath-hold. Informed consent was obtained from every subject.

### 2.2. MRI Protocols

A 3T machine (MAGNETOM Skyra, Siemens Healthcare, Erlangen, Germany) was used to acquire all MRCPs. An 18-channel body and 32-channel spine matrix coils were used for signal reception. The main advantage of 3T machines is a better CNR, which decreases the acquisition time and improves spatial resolution [22]. A recent study comparing 1.5T and 3T machines showed that image quality was better with CS than with the conventional sequence using the 3T machine, but that image quality was better with the conventional sequence using the 1.5T machine [24]. The patients started fasting 4 h before the scan; they were positioned supine and entered into the tunnel feet first. Before entering the tunnel, patients without diabetes drank a glass of pineapple juice, whose high manganese content decreases the T2 relaxation time of fluids, thereby suppressing the signal from the proximal digestive structures and limiting background noise [25]. SPACE was used to acquire 3D images. Conventional free-breathing with a navigator triggering sequence (RT-MRCP), that is the standard 3D-MRCP sequence used in our department, and the prototype BH sequence with compressed sensing (CS-BH-MRCP), were acquired according to the acquisition parameters shown in Table 1. For each MRCP sequence, maximum-intensity projection (MIP) images were generated automatically and used in addition of the native images for image analysis. The standard MRCP protocol also included a coronal and axial T2 Half-Fourier Acquisition Single-Shot Turbo Spin Echo (HASTE) sequence, as well as an axial T1 Volumetric Interpolated BH Examination (VIBE) with fat-saturation sequence.

RT-MRCP acquisition times were recorded. The time required for CS-BH-MRCP sequence image reconstruction on the MRI console was about 10 min. No other sequence could be acquired during reconstruction, and this time was therefore used to remove the patient from the machine and to prepare the next patient. Consequently, image quality could not be evaluated before the patient had left the MRI machine.

### 2.3. Image Quality Analysis

All quantitative and qualitative image analyses were performed on an image processing console with picture archiving and communication system (PACS; Centricity, GE Healthcare, Chicago, IL, USA).

#### 2.3.1. Quantitative Image Evaluation

Conventional methods based on the background region-of-interest (ROI), that are commonly used for signal-to-noise ratio and CNR assessment [12], might be unreliable with sequences using undersampling methods, such as PI and CS, due to the heterogeneous signal intensity of the background [26,27,28]. As it has been previously reported, the relative CBD to peri-biliary tissues (PBT) contrast ratio (CR) was assessed instead of the conventional SNR and CNR [23].

For each patient and each sequence, a representative section of the CBD was selected. Then, circular regions of interest (ROIs) were traced on the CBD and PBT and the mean signal intensities were recorded. The CBD ROI was at least 5 mm² and was placed in a uniform artifact-free region in the middle of the duct. A similar ROI was positioned on the peri-CBD tissues, avoiding artifacts and fluid-containing structures. Note that the ROIs were drawn on the native images. Figure 1 shows the ROIs placement. The CR was estimated using the following formula [23,29,30]:CR = (SI_CBD_ − SI_PBT_)/(SI_CBD_ + SI_PBT_)(1)
where CR is the estimated contrast ratio, SI_CBD_ and SI_PBT_ are the mean signal intensities of the CBD and the PBT, respectively.

#### 2.3.2. Qualitative Image Evaluation and Diagnostic Confidence

The images of the two MRCP sequences were read by two observers working independently of each other, a senior radiologist and a junior radiologist with 9 and 4 years of experience reading MRCPs, respectively. The readers were aware of the reason for per-forming MRCP but not of the MRCP protocol used.

Both the native images and the MIP images were used for the analysis. The image-processing console enabled multiplanar reconstructions. The other sequences from the standard protocol previously cited were also systematically reviewed.

Overall image quality, presence of artefacts, and background noise suppression were evaluated on a five-point Likert scale (Table 2). When artefacts were seen, their type was recorded as blurring artefacts, metal-induced artefacts, or wrap-around artefacts.

The readers also assessed the visualization of 12 segments of the pancreatic and bile ducts: the proximal and distal parts of the CBD; the confluence of the cystic duct and CBD; the proximal, central, and distal parts of the MPD; the intrahepatic bile ducts (IHBDs) to their primary branches (right and left IHBDs) and secondary branches (right anterior and posterior sectoral ducts and left medial and lateral branches). Each of these 12 segments were evaluated using a five-point Likert scale, with higher scores indicating better duct visualization (Table 2). Figure 2 shows these segments on an MIP reconstruction of a CS-BH-MRCP image.

In addition, adequate visualization of the entire biliary system as scores of 3 or more for the CBD, cystic duct confluence, and intrahepatic bile ducts to their primary and secondary branches, was defined. Similarly, adequate entire MPD visualization was de-fined as a score of 3 or more for all three segments of the MPD.

Degree of diagnostic confidence was scored on a 1–3 scale, with higher scores indicating greater confidence for the presence or absence of: CBD lithiasis, cyst communication with the MPD, biliary anastomosis stenosis (liver-transplant recipients) (Table 3). When diagnostic confidence was moderate to definitive (score of 2 or 3), the readers indicated whether CBD lithiasis, cyst communication with the MPD or biliary anastomosis stenosis (liver-transplant recipients) was seen (yes/no). In patients with more than one pancreatic cyst, communication with the MPD was evaluated only for the largest cyst.

CBD lithiasis, pancreatic cyst communication with the MDP and biliary anastomosis stenosis detection, based on the consensus between the two readers (confidence diagnostic of 2 or 3) with both sequences was reported. When available, the findings from ERCP for cholelithiasis were also reported; however, they did not serve as the reference standard, as lithiasis migration might have occurred during the interval between MRCP and ERCP.

### 2.4. Statistical Analysis

Variable values (CR) measured on the RT-MRCP sequence and the CS-BH-MRCT sequence were compared using Wilcoxon’s test. Overall image quality, presence of artefacts, background noise suppression, and duct visualization were described as mean ± standard deviation (SD) and range (minimum–maximum). Overall image quality and presence of artefacts for different RT-MRCP acquisition times were compared by applying the Kruskal–Wallis test. Inter-observer agreement was assessed by computing Cohen’s κ, which can vary from 0 to 1 (Appendix A, Table A1). The chi-square test was applied to compare visualization of the overall bile duct system and entire MPD. Bowker’s test was chosen to compare diagnostic confidence between the two sequences. MacNemar’s test was chosen to compare CBD lithiasis detection between the two sequences. Values of *p* smaller than 0.05 were taken to indicate significant between-group differences. Statistical analyses were performed using Stata 14.0 software (StataCorp, College Station, TX, USA).

## 3. Results

### 3.1. Study Population

Of 70 patients who met our inclusion criteria, two were unable to maintain a breath-hold, leaving 68 patients for the analysis. Table 4 lists the main patient features. Among them, 12 had history of cholecystectomy. Among the 54 patients referred for suspected choledocholithiasis, all presented liver function test abnormalities, 11 suffered from acute pancreatitis, and 10 suffered from acute cholecystitis. Teen patients were referred for pancreatic cyst characterization with suspected BD-IPMN and four liver-transplant patients were scanned for suspected stenosis of the biliary anastomosis.

### 3.2. Quantitative Image Evaluation

Concerning CBD and PBT CRs, a significant difference was found between the two sequences (*p* < 0.0001), that favored the CS-BH-MRCP sequence (Table 5).

### 3.3. Qualitative Image Evaluation

Table 6 compares the mean values of each variable regarding qualitative evaluation obtained by the two readers for each of the two sequences. Appendix A, Table A2 reports the results of the qualitative evaluation of the two MRCP sequences by each of the two readers. Interobserver agreement for the assessment of image quality qualitative evaluation variables ranged from moderate (κ = 0.41–0.60) to nearly perfect (κ = 0.81–1.00) (Appendix A, Table A3).

#### 3.3.1. Overall Image Quality, Artefacts and Background Noise Suppression

The CS-BH-MRCP sequence was associated with significantly better image quality (*p* = 0.004), significantly fewer artefacts (*p* = 0.004), and significantly better back-ground noise suppression (*p* = 0.011) overall. Blurring artefacts occurred with both sequences but were more numerous with the free-breathing sequence. Metal-induced artefacts were seen in two patients: For the first patient, the artefact was due to spinal internal fixation material and occurred on the free-breathing sequence and, for the second, the artefact was due to a weighted nasogastric feeding tube inserted as part of the management of acute pancreatitis and occurred on the BH sequence.

#### 3.3.2. Bile Ducts Visualization

Whereas visualization of the distal CBD was significantly better with the CS-BH-MRCP sequence (*p* = 0.015), no significant difference was found for the proximal CBD or cystic duct confluence (*p* = 0.054 and 0.459, respectively). Visualization of the right and left primary IHBD was significantly better with the CS-BH-MRCP sequence (*p* = 0.022 and 0.018, respectively), whereas no significant difference was found for the secondary IHBDs (Table 6).

#### 3.3.3. Main Pancreatic Duct Visualization

For the distal and proximal parts of the MPD, visualization was significantly better with the CS-BH-MRCP sequence (*p* = 0.001 and 0.032, respectively), whereas no difference was found for the central part (*p* = 0.885). The central part was often not seen on the CS-BH-MRCP sequence, as it was outside the acquisition volume.

#### 3.3.4. Entire Biliary System and Pancreatic Duct Visualization

Visualization of the entire biliary system was obtained in 48.5% (33/68) of patients with the RT-MRCP sequence and 35.3% (24/68) of patients with the CS-BH-MRCP se-quence; the difference was not significant (*p* = 0.118). Visualization of the entire MPD was obtained in 35.3% (24/68) of patients with the RT-MRCP sequence and in 51.5% (35/68) of patients with the CS-BH-MRCP sequence, with no significant difference (*p* = 0.057).

Figure 3 shows example MRCP images obtained with RT-MRCP and CS-BH-MRCP sequences with their qualitative image-quality criteria scorings.

#### 3.3.5. Acquisition Time and Overall Image Quality and Artefacts

The acquisition time was 17 s for all patients with the CS-BH-MRCP sequence. With the RT-MRCP sequence, the acquisition time ranged from 156 s (2 min 36 s) to 881 s (14 min 41 s). The mean time was 321 s, i.e., 5 min 21 s. Thus, the BH sequence was 19 times faster on average than the free-breathing sequence. Neither overall image quality nor presence of artefacts was associated with the RT-MRCP acquisition time (*p* = 0.458 and 0.250, respectively) (Appendix A, Table A4).

### 3.4. Diagnostic Confidence

Overall, diagnostic confidence was significantly better with the CS-BH-MRCP sequence for both readers (*p* = 0.038 for senior and *p* = 0.038 for junior, Appendix A, Table A5). Cross-tabulation analyses in Table 7 shows the diagnostic confidence scores obtained with CS-BH-MRCP versus RT-MRCP for both senior and junior radiologist. For the senior radiologist, diagnostic confidence was better with the CS-BH-MRCP sequence for 17 patients (25%) and with the RT-MRCP sequence for only 7 patients (10.3%). For the junior radiologist, the corresponding proportions were 21 (30.9%) and 9 (13.2%). Interobserver agreement for the assessment of the confidence diagnostic was moderate with RT-MRCP (κ = 0.59, 95% CI: 0.43–0.76) and substantial with CS-BH-MRCP (κ = 0.66, 95% CI: 0.50–0.81) (Appendix A, Table A3).

### 3.5. Bile Duct Lithiasis, Branch Duct Intraductal Papillary and Mucinous Neoplasms and Bile Duct Anastomosis Stenosis

Based on the consensus between the two readers, CBD lithiasis was present in 10 of the 54 patients. In three patients, the calculi were visible on both MRCP sequences and on a non-MRCP sequence and confirmed by ERCP. In five patients, CBD lithiasis was visible on the RT-MRCP sequence and on a non-MRCP sequence but not on the CS-BH-MRCP sequence. Among these five patients, only one had calculi visible by ERCP and three had uninterpretable CS-BH-MRCP images (diagnostic confidence of 1). Finally, in two patients, calculi were visible on the CS-BH-MRCP sequence and on non-MRCP sequences but not on the RT-MRCP sequence. Among these two patients, only one had calculi visible by ERCP. The RT-MRCP was considered interpretable in both patients. No significant difference was found between the CS-BH-MRCT and RT-MRCP sequences for the detection of bile-duct lithiasis (*p* = 0.30). One stone was fortuitously discovered in one patient referred for BD-IPMN, only on CS-BH-MRCP sequence (not included in analysis). Figure 4 shows three examples of MRCP images of patients referred for suspected choledocholithiasis using RT-MRCP and CS-BH-MRCP sequences. Appendix A, Table A6 shows the number of CBD lithiasis detection based on the consensus between the two readers with each MRCP sequence and the agreement between the two sequences.

Among the 10 patients who were scanned for suspected BD-IPMN, RT-MRCP images were considered uninterpretable (diagnostic confidence of 1) for four patients, whereas no CS-BH-MRCP images were considered uninterpretable. Communication between the pancreatic cyst and the MDP was visible in five patients with RT-MRCP sequence, whereas it was visible in six patients with CS-BH-MRCP sequence. Note that, regarding the cysts other imaging features, all lesions were most likely BD-IPMN, despite the inability to systematically demonstrate a communication with the MDP. Appendix A, Table A7 shows the number of patients for whom pancreatic cyst communication was visible each MRCP sequence and the agreement between the two sequences. Among the four patients with liver-transplant recipients who were referred for suspicion of biliary anastomosis stenosis, RT-MRCP and CS-BH-MRCP images were interpretable for all patients (diagnostic confidence of 2 or 3) and none of them presented noticeable stenosis. For the diagnostic of BD-IPMN and the detection of bile-duct anastomosis stenosis in liver-transplant recipients, the sample sizes were too small to allow a meaningful statistical analysis.

## 4. Discussion

In our study, based on quantitative and qualitative evaluations, the CS-BH-MRCP sequence was preferred over the conventional RT-MRCP sequence in terms of image quality at 3T with the advantage of a much shorter acquisition time. The CBD to PBT CR was significantly better. The CS-BH-MRCP sequence demonstrated significantly better overall image quality, fewer artefacts, better background noise suppression, a better visualization of the distal CBD, of the right and left primary IHBD and the distal and proximal parts of the MPD. No significant difference was found regarding the following qualitative criteria: the visualization of the proximal CBD, cystic duct confluence, secondary IHBDs, the central part of the MPD, the entire biliary system and entire MPD.

Overall, our results are consistent with previous reports. Three studies found a significantly better overall image quality with the CS-BH-MRCP sequence at 3T [9,24,31]. Studies also showed a significantly better visualization of the CBD [9], the primary IHBD [9,24], and the cystic duct [24], with the CS-BH-MRCP sequence at 3T. Nevertheless, the literature seems to show a discrepancy concerning the visualization of the MDP. With 200 patients scanned at 3T, the study of Blaise et al. showed a significantly better visualization of the MPD [24], although no segmental analysis was performed. This is in contrast with Zhu et al.’s prospective study including 80 patients, which found a worse visualization of the MPD with a CS-BH-MRCP sequence in comparison with a conventional NT-MRCP sequence at 3T [10]. CS-BH-MRCP had thus lower diagnostic sensitivity [10]. This finding prompted the same investigators to conduct another study evaluating a modified CS-BH-MRCP sequence with a smaller field of view (FOV) and higher spatial resolution that achieved better visualization of the MPD and secondary IHBDs than the “original” CS-BH-MRCP. This modified protocol also showed higher sensitivity for detecting pancreatic duct abnormalities [11]. Another optimized CS-BH-MRCP sequence at 3T with decreased accelerator factor and a reduced FOV and matrix without changes in spatial resolution, was proposed by Song et al., and demonstrated comparable or even better image quality than conventional MRCP [32]. Overall, the MPD was also better visualized with the optimized sequence [32].

Although our study’s results and previous reports suggested the superiority of the CS-BH-MRCP sequence at 3T, the same was not observed at 1.5T in some studies [24,31]. Taron et al.’s study suggested a better overall image quality with the conventional NT-MRCP at 1.5T, although the results were not significant [31]. At 1.5T, in Blaise et al.’s study, the conventional RT-MRCP acquisition showed a significant superior overall image quality with better visualization of the biliopancreatic ducts, whereas only sharpness was improved with BH-CS-MRCP [24]. In a recent study, a short single BH CS-MRCP sequence, that allowed a reduced acquisition time of 8 s, demonstrated higher scores for image quality, duct sharpness and duct visualization than the conventional NT-MRCP, a CS-NT-MRCP, and a long single BH CS-MRCP (acquisition time of 17 s) sequences, the results being not always significant for all criteria and sequence to sequence comparison [33]. This highlights the potential superiority of the CS-BH-MRCP sequence, even at 1.5T and with an even shorter acquisition time.

Unlike the proximal and distal MDP that were more clearly visualized with the CS-BH-MRCP, no difference was found concerning the central MPD. The same was observed for the secondary IHBDs. It might be partially explained by the lower number of coronal slices acquired with the CS-BH-MRCP than with the RT-MRCP (64 vs. 120 slices), resulting in a smaller acquisition volume. The central MPD and secondary IHBDs were indeed less frequently imaged with the CS-BH-MRCP, resulting in a bad duct visualization score of 1. Most of the patients were referred for suspected choledocholithiasis, the field of view was thus most likely centered on the CBD with less care being taken to cover the other pancreato-biliary ducts. Great care is, therefore, required when choosing the 3D imaging volume position most appropriate for the suspected diagnosis. The radiologic technologist must also ensure that the acquisition covers as many ducts as technically possible.

A significant difference in the CBD to periductal tissues CR between the two sequences that favored the CS-CH-MRCP sequence was found in our study. Although the CBD to periductal CR values were similar to values previously reported by Seo et al., i.e., 0.92 ± 0.03 for MRCP with PI and 0.91 ± 0.03 for MRCP with PI and CS, in their study, the significant difference in CR favored the MRCP sequence without CS [23]. Note that, unlike our study, both sequences were acquired using free-breathing navigator-triggered method and the two sequences acquisition parameters differed only for the acceleration factor and repetition time [23]. In Song et al.’s study, a significantly better CBD to PTB tissues CR was achieved using an optimized BH-CS-MRCP sequence compared to conventional MRCP (0.99 ± 0.01 versus 0.94 ± 0.04, *p* < 0.001), with slightly higher CR values compared to our study for both sequence [32].

In our study, 3D MRCP with CS was successfully acquired during a 17 s BH in 68 patients, i.e., 97% of the original cohort of 70 patients. This result was obtained although the patients had required hospital admission and exhibited multiple comorbidities likely to cause greater difficulty with maintaining a BH compared to the general population. Our study cohort was thus representative of everyday practice. Similar success rates have been reported [10,11]. The 10 min period needed for image reconstruction precluded immediate evaluation of image quality with repeated acquisition or the performance of additional sequences if needed. This point is a limitation to the use of the BH sequence instead of the free-breathing sequence. We used the reconstruction time to prepare the next patient. However, with the recently marketed latest version of the MRI machine software, the reconstruction time is only 20 s, allowing for the fast evaluation of the image quality of the acquisition and its repetition if necessary.

In the present study, radiologist diagnostic confidence was significantly better with the CS-BH-MRCP sequence that is certainly linked to the image quality. These results tend to support the use of the CS-BH-MRCP sequence for the diagnosis of biliary and ductal pancreatic diseases. Image quality indeed needs to be good enough in order to make a diagnosis with a very high degree of certainty. However, according to the senior reader, image quality was optimal to ensure complete confidence in the diagnosis (diagnostic confidence score of 3) in only 11.8% of patients with the RT-MRCP sequence and 26.5% with the CS-BH-MRCP sequence. Moderate confidence (diagnostic confidence score of 2) was achieved in 69.1% with the RT-MRCP and in 57.4% with the CS-BH-MRCP.

ERCP was rarely performed in our study and was often delayed. Furthermore, the sometimes lengthy times between MRCP and ERCP might explain the discrepancy between their findings, since spontaneous migration of the stone to the digestive tract might occur before ERCP. In addition, for patients who underwent ERCP in other centers, the results were not always available. We were therefore unable to assess and compare the diagnostic performance of the MRCP sequences. However, our analysis did not show a significant difference for the detection of bile duct lithiasis between the two sequences. Of the 11 patients with detected lithiasis, 6 had stones visible on the CS-BH-MRCP sequence. Given the short acquisition time of this sequence, it would be of interest to determine the sensitivity of a second acquisition in the event of bad image quality on the first acquisition. Tokoro et al.’s study suggested that the addition of the CS-BH-MRCP to the conventional MRCP protocol at 3T added value to the MRCP examination, since the CS-BH-MRCP could compensate for the image deterioration of the RT-MRCP caused by motion artefacts, although the image quality of the CS-BH-MRCP was not better than the RT-MRCP [34].

We were unable to further analyze the data on pancreatic cystic lesions, due to the weak sample size of this subgroup and the absence of available ERCP results. Nonetheless, the BH sequence visualized the proximal and distal parts of the MPD more clearly compared to the free-breathing sequence. In addition, a communication between the pancreatic cyst and the MDP was slightly more often visualized with the CS-BH-sequence, allowing the diagnosis of BD-IPMN. Therefore, the BH sequence may be relevant for evaluating pancreatic duct disorders, provided the acquisition volume is well centered on the MPD. The optimized CS-BH-MRCP proposed by Song et al. showed very interesting results and significantly better demonstrated the communication between the pancreatic cyst and the MPD as compared to the conventional MRCP [32]. With 41 patients included for the evaluation BD-IPMN using MRCP at 1.5T, the short single BH CS-MRCP sequence at 1.5T proposed by Henninger et al. demonstrated significantly higher scores in all the diagnostic approach criteria (lesion conspicuity, confidence, communication) compared to the conventional NT-MRCP, a CS-NT-MRCP, and a long single BH CS-MRCP sequences [33]. CS-BH-MRCP sequences that are specifically optimized for pancreatic ducts diseases assessment, could therefore improve the diagnostic performance in this indication.

The strengths of this study included the prospective design, reading of the images by two observers, exhaustive analysis of image parameters, and large number of patients compared to the published reports. However, our study presented several limitations. First, it was a single-center study. Second, the image quality analysis was mainly subjective. Nevertheless, the analysis was performed by two readers and interobserver agreement for the assessment of image quality qualitative evaluation variables ranged from moderate to substantial. Third, the excessively small subgroup sizes of patients with pancreatic cyst and liver-transplant recipients did not provide us enough data to allow a meaningful statistical analysis. Fourth, ERCP was rarely performed to confirm the diagnosis or the results were not available. We were therefore unable to assess the diagnostic performance of the MRCP sequences. However, this study mainly focused on image quality assessment. Fifth, blinded reading of the image parameters was biased by the recognizable appearance of CS-BH-MRCP images. Sixth, many acquisition parameters differed between the two sequences, that may make comparison challenging. However, this study compared a RT sequence with a BH-CS sequence. These are fundamentally different scan procedures that cannot be performed with identical protocol parameters. As such, these differences are not a true limitation of the study design but an inherent consequence of the applied techniques.

## 5. Conclusions

The present study shows that the CS-BH-MRCP sequence provides overall better image quality and bile and pancreatic ducts visualization compared to the conventional RT-MRCP sequence at 3T, with the advantage of a much shorter acquisition time. More studies are required to determine the diagnostic performance of this sequence for pancreato-biliary pathologies.

## Figures and Tables

**Figure 1 diagnostics-11-01886-f001:**
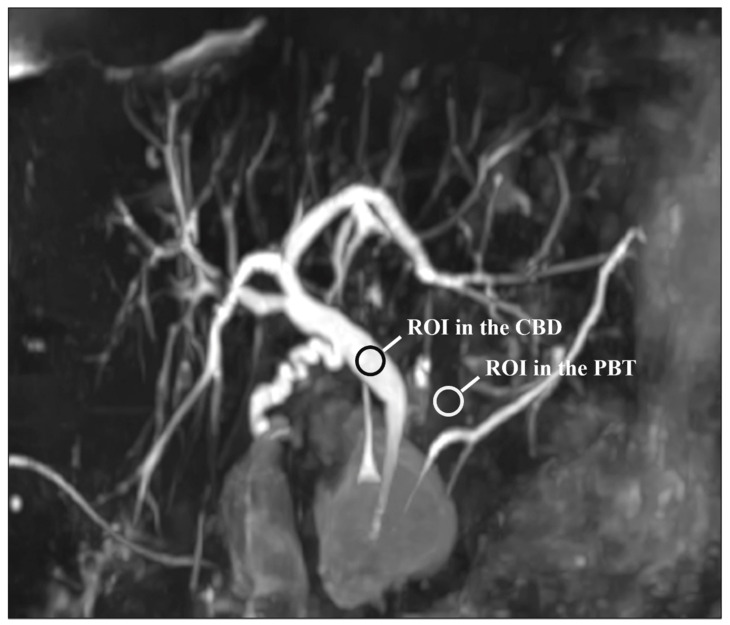
Representative section of the common bile duct (CBD) used for the quantitative evaluation of image quality: note the regions of interest (ROIs) on the CBD and peri-biliary tissues (PBT). Note that the ROIs were drawn on the native images and not on the MIP reconstruction as shown here.

**Figure 2 diagnostics-11-01886-f002:**
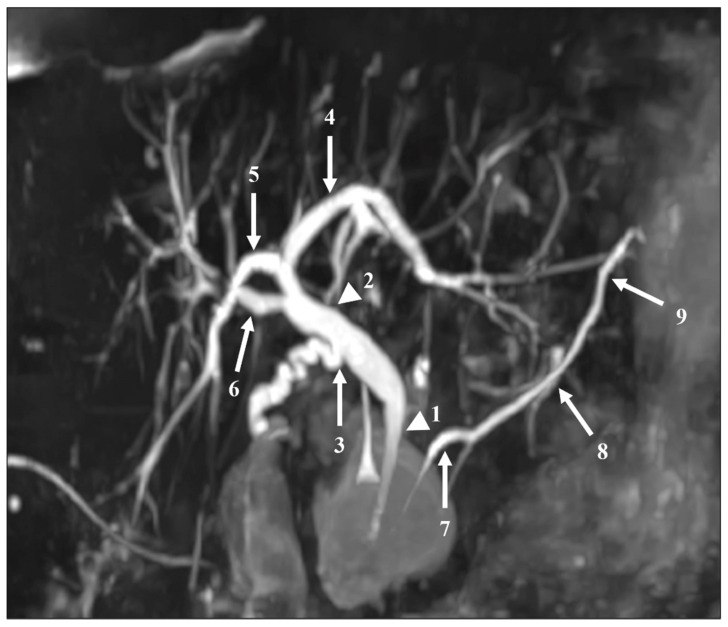
The segments of the pancreatic and bile ducts on a MIP reconstruction image acquired with the CS-BH-MRCP sequence for qualitative image quality analysis. Numbers represent the main pancreatic duct (MPD) and bile ducts: (1) distal common bile duct (CBD), (2) proximal CBD, (3) cystic duct confluence, (4) left primary intrahepatic bile duct (IBD), (5) right posterior sectoral IBD, (6) right anterior sectoral IBD, (7) distal MPD, (8) central MPD, (9) proximal MPD. Left medial and left lateral ducts are not represented on this figure. Note that no right primary IBD duct was present in this patient.

**Figure 3 diagnostics-11-01886-f003:**
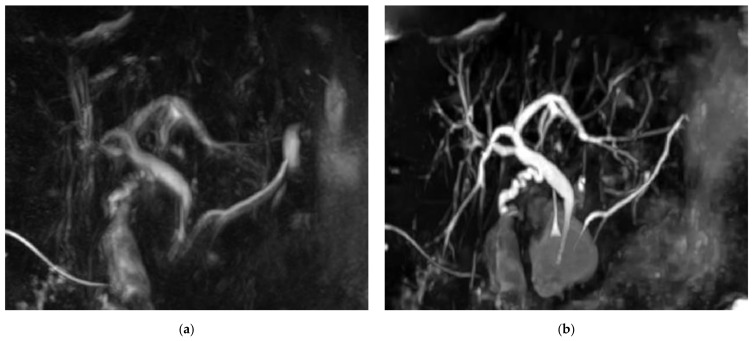
Example of MRCP images obtained with RT-MRCP and CS-BH-MRCP sequences: (**a**,**b**) MRCP was performed in this 69-year-old female with chronic cholecystitis for suspected choledocholithiasis. The motion artefacts noted on the RT-MRCP sequence (**a**) were less noticeable on the CS-BH-MRCP sequence (**b**). Note that qualitative image-quality criteria were assessed on native images, whereas MIP images are shown on this figure. (**a**) For RT-MRCP, qualitative image-quality criteria were scored by senior radiologist, as follows: overall image quality: 3; presence of artefacts: 3; background suppression: 3; distal CBP visualization: 3; proximal CBD visualization: 4; cystic duct confluence: 3; right primary IHBD visualization: 3; left primary IHBD: 3; right anterior sectoral duct: 2; right posterior sectoral duct: 2; left medial duct: 2; left lateral duct: 3; distal and central MDP: 3; proximal MDP: 2. (**b**) For CS-BH-MRCP qualitative image-quality criteria were scored by senior radiologist as follows: overall image quality: 5; presence of artefacts: 4; background suppression: 4; distal CBP visualization: 4; proximal CBD visualization: 5; cystic duct confluence: 5; right primary IHBD visualization: 5; left primary IHBD: 5; right anterior sectoral duct: 5; right posterior sectoral duct: 5; left medial duct: 5; left lateral duct: 5; distal and central MDP: 5; proximal MDP: 4. RT-MRCP acquisition time was 5 min 5 s.

**Figure 4 diagnostics-11-01886-f004:**
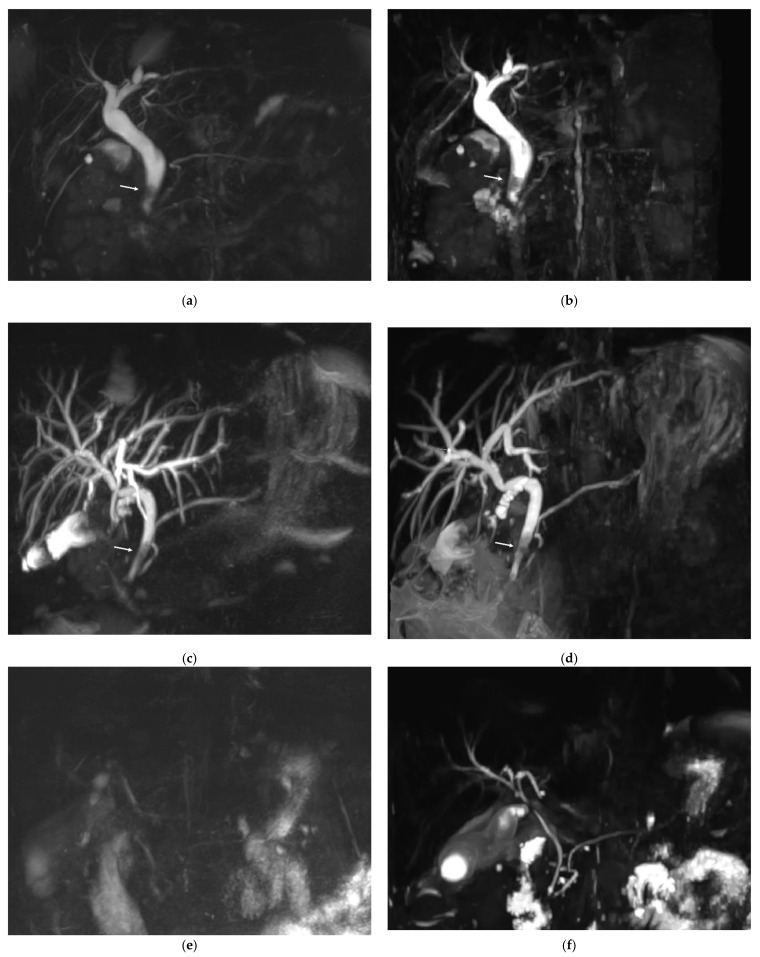
Three examples of MRCP images of patients referred for suspected choledocholithiasis using RT-MRCP and CS-BH-MRCP sequences. (**a**,**b**) MRCP images from a 41-year-old female. The stone within the common bile duct (arrow) was clearly seen with both RT-MRCP (**a**) and CS-BH-MRCP (**b**) sequences with a diagnostic confidence score of 3. RT-MRCP acquisition time was 4 min 30 s. (**c**,**d**) MRCP images from a 61-year-old female. Both RT-MRCP (**c**) and CS-BH-MRCP (**d**) sequences showed intra-hepatic bile duct dilation upstream of calculi (arrows) in the common bile duct with a diagnostic confidence score of 3. RT-MRCP acquisition time was 3 min 23 s. (**e**,**f**) MRCP images from a 55-year-old male. (**e**) RT-MRCP showed very poor image quality and was uninterpretable. The diagnostic confidence was scored 1. (**f**) CS-BH-MRCP showed better image quality, with a diagnostic confidence score of 2, allowing moderate confidence to diagnose the absence of CBD lithiasis. RT-MRCP acquisition time was 14 min 41 s.

**Table 1 diagnostics-11-01886-t001:** Acquisition parameters of the conventional RT-MRCP and prototype CS-BH-MRCP sequences.

Parameters	RT-MRCP	CS-BH-MRCP
TR (ms)	Variable, dependent on breathing rate	1700
TE (ms)	629	426
Flip angle (degrees)	115	115
Matrix	346 × 384	307 × 384
Acquired voxel size (mm^3^)	1.13 × 1.02 × 2.03	1.30 × 1.04 × 2.20
Reconstructed voxel size (mm^3^)	0.51 × 0.51 × 1.30	0.52 × 0.52 × 1.10
Acceleration factor	GRAPPA: 2	23 *
FOV (mm^2^)	390 × 390	400 × 400
Number of coronal slices	120	64
Number of averages	1.4	1.4
Oversampling phase	30%	0%
Oversampling slice	6.70%	200%
Reference lines	24	24
Turbo factor	180	207
Echo-train duration (ms)	895	823
Bandwidth (Hz/pixel)	352	501
Echo-spacing (ms)	4.92	3.94
Acquisition time (s)	Mean: 321 (range: 156–881)	17

RT-MRCP, respiratory-triggered magnetic resonance cholangiopancreatography; CS-BH-MRCP, compressed sensing breath-hold magnetic resonance cholangiopancreatography; TR, repetition time; TE, echo time; FOV, field of view; * k-space was incoherently undersampled allowing acceleration factor of 23.

**Table 2 diagnostics-11-01886-t002:** Likert scale used to evaluate the qualitative image-quality: overall image quality, presence of artefacts, background suppression, and bile-duct visualization.

Scores	Overall Image Quality	Presence of Artefacts	Background Suppression	Duct Visualization
1	No diagnosis can be made	Severe artefacts precluding a diagnosis	Significant background noise precluding image interpretation	Structure not visible
2	Image quality too poor to allow a diagnosis	Major artefacts severely impeding the ability to make a diagnosis	Notable background noise raising major challenges with image interpretation	Vague visualization of a duct-like structure
3	Acceptable image quality allowing a diagnosis	Moderate artefacts making the diagnosis uncertain	Moderate background noise that hinders interpretation	Partially visible duct
4	Good image quality allowing a diagnosis	Minor artefacts that do not preclude a diagnosis	Minimal background noise that does not hinder the interpretation of the bile-duct images	Most of the structure is visible, with some blurriness
5	Excellent image quality allowing a diagnosis with a good degree of confidence	Excellent image quality with no artefacts	Excellent background noise suppression	The entire duct is seen clearly

**Table 3 diagnostics-11-01886-t003:** Likert scale used to evaluate the degree of confidence in the diagnosis.

Score	Degree of DC	Presence or Absence of		
		CBD Lithiasis	Cyst Communication with the MPD	Biliary Anastomosis Stenosis (LTR)
1	No confidence, no diagnosis established	Could not be determined		
2	Moderate confidence, diagnosis probable	Doubtful but possible		
3	Complete confidence, definitive diagnosis	Certainly present		

DC, diagnostic confidence; CBD, common bile duct; MPD, main pancreatic duct; LTR, liver-transplant recipients.

**Table 4 diagnostics-11-01886-t004:** Main features of the 68 study patients.

Features	Mean ± SD/No. (%)
Age (years)	61.0 ± 15.1
Males/Females	26 (38.2%)/42 (61.8%)
Height (m)	1.7 ± 0.1
Weight (kg)	73.9 ± 17.2
Body mass index (kg/m²)	26.7 ± 5.6
History of surgery	
Prior cholecystectomy	12 (17.6%)
Liver transplant recipient	4 (5.8%)
Reason for MRCP	
Suspected choledocholithiasis	54 (79.4%)
Suspected BD-IPMN	10 (14.7%)
Suspected bile-duct stenosis after LT	4 (5.8%)

SD, standard deviation; No., number; BD-IPMN, branch-duct intraductal papillary mucinous neoplasm; LT, liver transplantation.

**Table 5 diagnostics-11-01886-t005:** Quantitative evaluation: comparison of the common bile duct (CBD) to peri-biliary tissues (PBT) contrast ratios (CR) between the two sequences RT-MRCP and CS-BH-MRCP.

	RT-MRCP	CS-BH-MRCP	*p* Value
CBD/PBT CR			<0.0001
Mean ± SD	0.894 ± 0.067	0.957 ± 0.047	
Median (range *)	0.911 (0.697–0.976)	0.974 (0.747–1.000)	

RT-MRCP, respiratory-triggered magnetic resonance cholangiopancreatography; CS-BH-MRCP, compressed sensing breath-hold magnetic resonance cholangiopancreatography; CR, contrast ratio; CR = (SI_CBD_ − SI_PBT_)/(SI_CBD_ + SI_PBT_); CBD, common bile duct; PBT, peri-biliary tissues; * range: minimum-maximum values.

**Table 6 diagnostics-11-01886-t006:** Qualitative evaluation: Comparison of the two sequences RT-MRCP and CS-BH-MRCP: mean value [mean ± SD, (range)] of each qualitative variable based a 5-point Likert scale, obtained by the two readers.

Variables	RT-MRCP	CS-BH-MRCP	*p* Value
Overall image quality	3.2 ± 0.9 (1–5)	3.5 ± 1.0 (1–5)	0.004
Presence of artefacts	3.2 ± 0.8 (1–4.5)	3.5 ± 0.9 (1–5)	0.004
Background noise suppression	3.4 ± 0.8 (1.5–4.5)	3.7 ± 0.9 (1.5–5)	0.011
Visualization of the BD			
Common bile duct			
*CBD, distal*	3.7 ± 0.8 (1–5)	4.0 ± 0.9 (1–5)	0.015
*CBD, proximal*	3.9 ± 0.7 (1–5)	4.1 ± 0.8 (1–5)	0.054
Cystic duct confluence	3.2 ± 1.3 (1–5)	3.3 ± 1.5 (1–5)	0.459
Primary IHBDs			
*Right primary IHBD*	3.7 ± 1.0 (1–5)	4.0 ± 0.9 (1–5)	0.022
*Left primary IHBD*	3.6 ± 1.1 (1–5)	4.0 ± 0.9 (1–5)	0.018
Secondary IHBDs			
*Right anterior sectoral duct*	3.2 ± 1.2 (1–5)	3.3 ± 1.2 (1–5)	0.463
*Right posterior sectoral duct*	3.2 ± 1.2 (1–5)	3.2 ± 1.2 (1–5)	0.972
*Left medial duct*	3.0 ± 1.3 (1–5)	3.0 ± 1.3 (1–5)	0.901
*Left lateral duct*	2.9 ± 1.3 (1–5)	2.8 ± 1.4 (1–5)	0.766
Visualization of the MPD			
MPD, distal	3.3 ± 1.1 (1–5)	3.7 ± 1.1 (1–5)	0.001
MPD, central	3.2 ± 1.1 (1–5)	3.2 ± 1.4 (1–5)	0.885
MPD, proximal	2.4 ± 1.1 (1–5)	2.8 ± 1.3 (1–5)	0.032

RT-MRCP, respiratory-triggered magnetic resonance cholangiopancreatography; CS-BH-MRCP, compressed sensing breath-hold magnetic resonance cholangiopancreatography; BD, bile ducts; CBD, common bile duct; IHBDs, intrahepatic bile ducts; MPD, main pancreatic duct.

**Table 7 diagnostics-11-01886-t007:** Cross-tabulation analyses: diagnostics confidence scores with CS-BH-MRCP versus RT-MRCP sequences for senior and junior readers.

		CS-BH-MRCP			
	Diagnostic Confidence Scores	1	2	3	Total
RT-MRCP	Senior				
	1	5 (7.4%)	6 (8.8%)	2 (2.9%)	13 (19.1%)
	2	6 (8.8%)	32 (47.1%)	9 (13.2%)	47 (69.1%)
	3	0 (0.0%)	1 (1.5%)	7 (10.3%)	8 (11.8%)
	Total	11 (16.2%)	39 (57.4%)	18 (26.5%)	68 (100%)
	Junior				
	1	3 (4.4%)	6 (8.8%)	3 (4.4%)	12 (17.7%)
	2	6 (8.8%)	17 (25.0%)	12 (17.7%)	35 (51.5%)
	3	0 (0.0%)	3 (4.4%)	18 (26.5%)	21 (30.9%)
	Total	9 (13.2%)	26 (38.2%)	33 (48.5%)	68 (100%)

RT-MRCP, respiratory-triggered magnetic resonance cholangiopancreatography; CS-BH-MRCP, compressed sensing breath-hold magnetic resonance cholangiopancreatography; green color: number of patients with a better diagnostic confidence score with the CS-BH-MRCP sequence; blue color: number of patients with a better diagnostic confidence score with the RT-MRCP sequence.

## Data Availability

The data presented in this study are available on request from the corresponding author. The data are not publicly available due to identity reasons.

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
