# Peer review of "Single-Breath-Hold MRI-SPACE Cholangiopancreatography with Compressed Sensing versus Conventional Respiratory-Triggered MRI-SPACE Cholangiopancreatography at 3Tesla: Comparison of Image Quality and Diagnostic Confidence"

_diagnostics, 2021, doi:10.3390/diagnostics11101886_

Round 1

Reviewer 1 Report

This manuscript compared a prototype compressed-sensing breath-hold SPACE sequence (CS-BH-MRCP) with a traditional sequence for image quality and diagnostic confidence on Cholangiopancreatography. The design of study is scientifically solid and convincing. The sample size included in the study was decent. One thing I wish the authors has done was including larger number of radiologists to read the images and compare the discrepancy among them. Overall, the result and conclusion of the study is persuasive.

I have few comments as below:

Comment 1.

Page 4, Table 1

Can the authors list the range of acquisition time for RT-MRCP to compare with BH-MRCP? I understand the acquisition time with a RT sequence is highly variable patient to patient, but it is nice to know the range and average that allows a direct comparison with BH-MRCP. Or it can be in the notes under the table, line 136 to line 140.

Comment 2

Page 4, line 152-157

The authors described in detail the selection of ROI of CBD and ROI of PBT. And they mentioned” The CBD ROI was at least 5 mm² and was placed in a uniform artifact-free region in the middle of the duct.”

Just curious on the selection of region on the CBD, anatomically is relative small and narrow region, if the ROI was at least 5 mm2, which is around 2.5 mm in diameter considering the ROI as round shape. In the acquisition parameter tables for the CS-BH-MRCP sequence, the matrix is 307 by 384, the FOV is 400 by 400. Therefore, one pixel is sized around 1.3 by 1.04 mm. Was the ROI too small and make the selection not accurate in the study?

Author Response

Responses to Reviewer 1 Comments

This manuscript compared a prototype compressed-sensing breath-hold SPACE sequence (CS-BH-MRCP) with a traditional sequence for image quality and diagnostic confidence on Cholangiopancreatography. The design of study is scientifically solid and convincing. The sample size included in the study was decent. One thing I wish the authors has done was including larger number of radiologists to read the images and compare the discrepancy among them. Overall, the result and conclusion of the study is persuasive.

Reply: Thank you very much for your comments.

I have few comments as below:

Comment 1.

Page 4, Table 1

Can the authors list the range of acquisition time for RT-MRCP to compare with BH-MRCP? I understand the acquisition time with a RT sequence is highly variable patient to patient, but it is nice to know the range and average that allows a direct comparison with BH-MRCP. Or it can be in the notes under the table, line 136 to line 140.

Reply: Thank you very much for your comments. As suggested, the mean acquisition time and time values range for RT-MRCP are shown in lines 314-315, paragraph 3.3.5, as follows: “With the RT-MRCP sequence, the acquisition time ranged from 156 s (2 min 36 s) to 881 s (14 min 41 s). The mean time was 321 s, i.e., 5 min 21 s. Thus, the BH sequence was 19 times faster on average than the free-breathing sequence.” Following your comment, the mean acquisition time (and range) was added in Table 1. In addition, we noticed that the BH sequence acquisition time was wrong in Table 1: 14s instead of 17s. The acquisition time was 17s as stated in paragraph 3.3.5 and the discussion part (and in table 1 with TR: 1700 ms). We made the corrections accordingly.

Comment 2

Page 4, line 152-157

The authors described in detail the selection of ROI of CBD and ROI of PBT. And they mentioned” The CBD ROI was at least 5 mm² and was placed in a uniform artifact-free region in the middle of the duct.”

Just curious on the selection of region on the CBD, anatomically is relative small and narrow region, if the ROI was at least 5 mm2, which is around 2.5 mmin diameter considering the ROI as round shape. In the acquisition parameter tables for the CS-BH-MRCP sequence, the matrix is 307 by 384, the FOV is 400 by 400. Therefore, one pixel is sized around 1.3 by 1.04 mm. Was the ROI too small and make the selection not accurate in the study?

Reply: Thank you very much for your comment. The ROI was traced approximately on the middle part of the CBD. Indeed, this region can be quite narrow, the CBD diameters at the ROI level were recorded and ranged from 3 mm (for only 1 patient) to 14 mm. For only 8 patients the CBD diameter was < 5 mm. We recognize that for these few patients the ROI was not strictly circular and was oval-shaped.

We also acknowledge that a ROI of 5 mm² contained only few pixels. However, the ROI was at least 5 mm² (i.e. for only very few patients) and was generally quite larger. Unfortunately, the ROIs sizes were not recorded. Minimum ROI size of 5 mm² in bile duct was already reported in several studies as follows:

Lee, J.H.; Lee, S.S.; Kim, J.Y.; Kim, I.S.; Byun, J.H.; Park, S.H.; Lee, M.G. Parallel imaging improves the image quality and duct visibility of breathhold two-dimensional thick-slab MR cholangiopancreatography. J. Magn. Reson. Imaging 2014, 39, 269–275. doi:10.1002/jmri.24155.

Seo, N.; Park, M.S.; Han, K.; Kim, D.; King, K.F.; Choi, J.Y.; Kim, H.; Kim, H.J.; Lee, M.; Bae, H.; Kim, M.J. Feasibility of 3D navigator-triggered magnetic resonance cholangiopancreatography with combined parallel imaging and compressed sensing reconstruction at 3T. J. Magn. Reson. Imaging 2017, 46, 1289–1297. doi:10.1002/jmri.25672.

Reviewer 2 Report

Diagnostics-1407362

It is very well known that using randomised acquisition, compressed sensing significantly shortens the overall MR scan time and attains benefits from that, such as SNR (by more averages), resolution, motion artefact, etc. This manuscript demonstrates one of the good examples of using CS. Overall, this manuscript is well written and the results are promising. However, my main concern is associated with the scan protocol and comparison of results acquired using two different sequences.  

Obviously, the major scan parameters (i.e., TR and TE) are not identical which may provide the result biased. Even though it might be possible, MR contrasts substantially rely on these parameters. So with these different variables, we cannot conclude one is better than the other. Please also check the in-plane resolution, slice thickness, number of slices, oversampling, etc which could also influence the results. What is the overall acquisition time for the conventional sequence? This is perhaps the most important information (Just mentioning “Variable” cannot be accepted). Different turbo factors could also contribute to the acquisition time. What is the acceleration factor, which is also quite different between the two sequences? Is it GRAPPA factor?

It would be really great to create new contrast or to reveal a conventionally invisible part using the proposed sequence at the same field strength. But this is not the case here.  

Have you also considered measuring T2 values at various regions? What about the accuracy of T2 between two different sequences when it is compared to the gold standard?

Author Response

Responses to Reviewer 2 Comments

It is very well known that using randomised acquisition, compressed sensing significantly shortens the overall MR scan time and attains benefits from that, such as SNR (by more averages), resolution, motion artefact, etc. This manuscript demonstrates one of the good examples of using CS. Overall, this manuscript is well written and the results are promising. However, my main concern is associated with the scan protocol and comparison of results acquired using two different sequences.  

Reply: Thank you very much for your comments. A detailed response has been provided below regarding your concern.

Obviously, the major scan parameters (i.e., TR and TE) are not identical which may provide the result biased. Even though it might be possible, MR contrasts substantially rely on these parameters. So with these different variables, we cannot conclude one is better than the other. Please also check the in-plane resolution, slice thickness, number of slices, oversampling, etc which could also influence the results. What is the overall acquisition time for the conventional sequence? This is perhaps the most important information (Just mentioning “Variable” cannot be accepted). Different turbo factors could also contribute to the acquisition time. What is the acceleration factor, which is also quite different between the two sequences? Is it GRAPPA factor?

Authors: Thank you very much for your comments. The mean acquisition time and values range for RT-MRCP are shown lines 314-315, paragraph 3.3.5 as follows: “With the RT-MRCP sequence, the acquisition time ranged from 156 s (2 min 36 s) to 881 s (14 min 41 s). The mean time was 321 s, i.e., 5 min 21 s. Thus, the BH sequence was 19 times faster on average than the free-breathing sequence.” Following your comment, the mean and range values of RT-sequence acquisition time were added in Table 1.

In addition, we noticed that the BH sequence acquisition time was wrong in Table 1: 14s instead of 17s. The acquisition time was 17s as stated in paragraph 3.3.5 and the discussion part (and in table 1 with TR: 1700 ms). We made the corrections.

This study compared a RT-sequence with a breathhold sequence. These are fundamentally different scan procedures that cannot be performed with identical protocol parameters. The CS-acquisition is done in a breath hold, which poses strict restrictions to the scan time to be within a breath hold duration. In addition, the CS-acquisition technique requires parameter adaptation to address known limitations of the technique. In this CS protocol, e.g. extensive slice oversampling was used to address aliasing artifacts. Therefore, it is clinically not possible to run the CS approach with a similar parameterization as the clinical standard protocol, which would provide the ideal comparison of course. Protocol parameters in the CS protocol were changed to accelerate the protocol conventionally, e.g. shorter TR, longer TE. In combination with a reduced TR it cannot be guaranteed that the contrast is identical between the protocols. But even at a TE of 426 ms the signal of most other tissues is already decayed so that the fluid filled ducts are the dominant signal here as also seen for longer TE. It may be also noted that some papers, which also performed a comparison between conventional and CS MRCP protocols, also showed some variations of TE and other parameters between the protocols. E.g. [11]:

Zhu, L.; Xue, H.; Sun, Z.; Qian, T.; Weiland, E.; Kuehn, B.; Asbach, P.; Hamm, B.; Jin, Z. modified breath-hold compressed-sensing 3D MR cholangiopancreatography with a small field-of-view and high resolution acquisition: clinical feasibility in biliary and pancreatic disorders. J. Magn. Reson. Imaging 2018, 48, 1389–1399. doi:10.1002/jmri.26049

Resolution also varies between the protocols, which influence the visibility of fine tissue structures, but with less impact on contrast.

The acceleration factor of the conventional protocol is indeed the GRAPPA factor. For CS, k-space is incoherently undersampled, which enables such a high acceleration factor. Table 1 was changed accordingly: Grappa 2 has been added for RT-MRCP and we detailed the acceleration factor method in the notes under the table (*).

We believe that these differences in acquisition parameters are not a true limitation of the study design but an inherent consequence of the applied techniques.

The next sentences were added in the discussion part (lines 510 to 515): “Sixth, many acquisition parameters differed between the two sequences, that may make comparison challenging. However, this study compared a RT sequence with a BH-CS sequence. These are fundamentally different scan procedures that cannot be performed with identical protocol parameters. As such, these differences are not a true limitation of the study design but an inherent consequence of the applied techniques.”

It would be really great to create new contrast or to reveal a conventionally invisible part using the proposed sequence at the same field strength. But this is not the case here.   

Have you also considered measuring T2 values at various regions? What about the accuracy of T2 between two different sequences when it is compared to the gold standard?

Reply: Thank you very much for your comments. To our knowledge, T2 values of tissues can only be assessed by varying the TE, as it is done with T2-mapping sequence. We do not fully understand how to perform quantitative assessment using these sequences.

Round 2

Reviewer 2 Report

Thank you for replying to my comments. I have no further comments.